# On the Optimization Landscape of Tensor Decompositions

**Rong Ge**
Duke University
`rongge@cs.duke.edu`

**Tengyu Ma**
Facebook AI Research
`tengyuma@cs.stanford.edu`

## Abstract

Non-convex optimization with local search heuristics has been widely used in machine learning, achieving many state-of-art results. It becomes increasingly important to understand why they can work for these NP-hard problems on typical data. The landscape of many objective functions in learning has been conjectured to have the geometric property that "all local optima are (approximately) global optima", and thus they can be solved efficiently by local search algorithms. However, establishing such property can be very difficult.

In this paper, we analyze the optimization landscape of the random over-complete tensor decomposition problem, which has many applications in unsupervised leaning, especially in learning latent variable models. In practice, it can be efficiently solved by gradient ascent on a non-convex objective. We show that for any small constant $\varepsilon > 0$, among the set of points with function values $(1 + \varepsilon)$-factor larger than the expectation of the function, all the local maxima are approximate global maxima. Previously, the best-known result only characterizes the geometry in small neighborhoods around the true components. Our result implies that even with an initialization that is barely better than the random guess, the gradient ascent algorithm is guaranteed to solve this problem.

Our main technique uses Kac-Rice formula and random matrix theory. To our best knowledge, this is the first time when Kac-Rice formula is successfully applied to counting the number of local optima of a highly-structured random polynomial with dependent coefficients.

## 1 Introduction

Non-convex optimization is the dominating algorithmic technique behind many state-of-art results in machine learning, computer vision, natural language processing and reinforcement learning. Local search algorithms through stochastic gradient methods are simple, scalable and easy to implement. Surprisingly, they also return high-quality solutions for practical problems like training deep neural networks, which are NP-hard in the worst case. It has been conjectured [DPG+14, CHM+15] that on *typical* data, the landscape of the training objectives has the nice geometric property that *all local minima are (approximate) global minima*. Such property assures the local search algorithms to converge to global minima [GHJY15, LSJR16, NP06, SQW15]. However, establishing it for concrete problems can be challenging.

Despite recent progress on understanding the optimization landscape of various machine learning problems (see [GHJY15, BBV16, BNS16, Kaw16, GLM16, HM16, HMR16] and references therein), a comprehensive answer remains elusive. Moreover, all previous techniques fundamentally rely on the spectral structure of the problems. For example, in [GLM16] allows us to pin down the set of the critical points (points with vanishing gradients) as approximate eigenvectors of some matrix. Among

these eigenvectors we can further identify all the local minima. The heavy dependency on linear algebraic structure limits the generalization to problems with non-linearity (like neural networks).

Towards developing techniques beyond linear algebra, in this work, we investigate the optimization landscape of tensor decomposition problems. This is a clean non-convex optimization problem whose optimization landscape cannot be analyzed by the previous approach. It also connects to the training of neural networks with many shared properties [NPOV15] . For example, in comparison with the matrix case where all the global optima reside on a (connected) Grassmannian manifold, for both tensors and neural networks all the global optima are isolated from each other.

Besides the technical motivations above, tensor decomposition itself is also the key algorithmic tool for learning many latent variable models, mixture of Gaussians, hidden Markov models, dictionary learning [Cha96, MR06, HKZ12, AHK12, AFH$^+$12, HK13], just to name a few. In practice, local search heuristics such as alternating least squares [CLA09], gradient descent and power method [KM11] are popular and successful.

Tensor decomposition also connects to the learning of neural networks [GLM17, JSA15, CS16]. For example, The work [GLM17] shows that the objective of learning one-hidden-layer network is implicitly decomposing a sequence of tensors with shared components, and uses the intuition from tensor decomposition to design better objective functions that provably recovers the parameters under Gaussian inputs.

Concretely, we consider decomposing a random 4-th order tensor $T$ of the rank $n$ of the following form,

$$T = \sum_{i=1}^{n} a_i \otimes a_i \otimes a_i \otimes a_i \,.$$

We are mainly interested in the *over-complete* regime where $n \gg d$. This setting is particularly challenging, but it is crucial for unsupervised learning applications where the hidden representations have higher dimension than the data [AGMM15, DLCC07]. Previous algorithmic results either require access to high order tensors [BCMV14, GVX13], or use complicated techniques such as FOOBI [DLCC07] or sum-of-squares relaxation [BKS15, GM15, HSSS16, MSS16].

In the worst case, most tensor problems are NP-hard [Hås90, HL13]. Therefore we work in the average case where vectors $a_i \in \mathbb{R}^d$ are assumed to be drawn i.i.d from Gaussian distribution $\mathcal{N}(0, I)$. We call $a_i$'s the components of the tensor. We are given the entries of tensor $T$ and our goal is to recover the components $a_1, \ldots, a_n$.

We will analyze the following popular non-convex objective,

$$\max \quad f(x) = \sum_{i,j,k,l \in [d]^4} T_{i,j,k,l} x_i x_j x_k x_l = \sum_{i=1}^{n} \langle a_i, x \rangle^4 \tag{1.1}$$
$$s.t. \quad \|x\| = 1.$$

It is known that for $n \ll d^2$, the global maxima of $f$ is close to one of $\pm \frac{1}{\sqrt{d}} a_1, \ldots, \pm \frac{1}{\sqrt{d}} a_n$. Previously, Ge et al. [GHJY15] show that for the orthogonal case where $n \leq d$ and all the $a_i$'s are *orthogonal*, objective function $f(\cdot)$ have only $2n$ local maxima that are approximately $\pm \frac{1}{\sqrt{d}} a_1, \ldots, \pm \frac{1}{\sqrt{d}} a_n$. However, the technique heavily uses the orthogonality of the components and is not generalizable to over-complete case.

Empirically, projected gradient ascent and power methods find one of the components $a_i$'s even if $n$ is significantly larger than $d$. The local geometry for the over-complete case around the true components is known: in a small neighborhood of each of $\pm \frac{1}{\sqrt{d}} a_i$'s, there is a unique local maximum [AGJ15]. Algebraic geometry techniques [CS13, ASS15] can show that $f(\cdot)$ has an exponential number of other critical points, while these techniques seem difficult to extend to the characterization of local maxima. It remains a major open question whether there are any other spurious local maxima that gradient ascent can potentially converge to.

**Main results.** We show that there are no spurious local maxima in a large superlevel set that contains all the points with function values slightly larger than that of the random initialization.

**Theorem 1.1.** *Let $\varepsilon, \zeta \in (0, 1/3)$ be two arbitrary constants and $d$ be sufficiently large. Suppose $d^{1+\varepsilon} < n < d^{2-\varepsilon}$. Then, with high probability over the randomness of $a_i$'s, we have that in the superlevel set*

$$L = \left\{ x \in S^{d-1} : f(x) \geq 3(1+\zeta)n \right\}, \tag{1.2}$$

*there are exactly $2n$ local maxima with function values $(1 \pm o(1))d^2$, each of which is $\widetilde{O}(\sqrt{n/d^3})$-close to one of $\pm\frac{1}{\sqrt{d}}a_1, \dots, \pm\frac{1}{\sqrt{d}}a_n$.*

Previously, the best known result [AGJ15] only characterizes the geometry in small neighborhoods around the true components, that is, there exists one local maximum in each of the small constant neighborhoods around each of the true components $a_i$'s. (It turns out in such neighborhoods, the objective function is actually convex.) We significantly enlarge this region to the superlevel set $L$, on which the function $f$ is not convex and has an exponential number of saddle points, but still doesn't have any spurious local maximum.

Note that a random initialization $z$ on the unit sphere has expected function value $\mathbb{E}[f(z)] = 3n$. Therefore the superlevel set $L$ contains all points that have function values barely larger than that of the random guess. Hence, Theorem 1.1 implies that with a slightly better initialization than the random guess, gradient ascent and power method[1] are guaranteed to find one of the components in polynomial time. (It is known that after finding one component, it can be peeled off from the tensor and the same algorithm can be repeated to find all other components.)

**Corollary 1.2.** *In the setting of Theorem 1.1, with high probability over the choice of $a_i$'s, we have that given any starting point $x^0$ that satisfies $f(x^0) \geq 3(1+\zeta)n$, stochastic projected gradient descent[2] will find one of the $\pm\frac{1}{\sqrt{d}}a_i$'s up to $\widetilde{O}(\sqrt{n/d^3})$ Euclidean error in polynomial time.*

We also strengthen Theorem 1.1 and Corollary 1.2 (see Theorem 3.1) slightly – the same conclusion still holds with $\zeta = O(\sqrt{d/n})$ that is smaller than a constant. Note that the expected value of a random initialization is $3n$ and we only require an initialization that is slightly better than random guess in function value. We remark that a uniformly random point $x$ on the unit sphere are **not** in the set $L$ with high probability. It's an intriguing open question to characterize the landscape in the complement of the set $L$.

We also **conjecture** that from random initialization, it suffices to use constant number of projected gradient descent (with optimal step size) to achieve the function value $3(1+\zeta)n$ with $\zeta = O(\sqrt{d/n})$. This conjecture — an interesting question for future work — is based on the hypothesis that the first constant number of steps of gradient descent can make similar improvements as the first step does (which is equal to $c\sqrt{dn}$ for a universal constant $c$).

As a comparison, previous works such as [AGJ15] require an initialization with function value $\Theta(d^2) \gg n$. Anandkumar et al. [AGJ16] analyze the dynamics of tensor power method with a delicate initialization that is *independent* with the randomness of the tensor. Thus it is not suitable for the situation where the initialization comes from the result of another algorithm, and it does not have a direct implication on the landscape of $f(\cdot)$.

We note that the local maximum of $f(\cdot)$ corresponds to the robust eigenvector of the tensor. Using this language, our theorem says that a robust eigenvector of an over-complete tensor with random components is either one of those true components or has a small correlation with the tensor in the sense that $\langle T, x^{\otimes 4} \rangle$ is small. This improves significantly upon the understanding of robust eigenvectors [ASS15] under an interesting random model.

The condition $n > d^{1+\varepsilon}$ should be artificial. The under-complete case ($n < d$) can be proved by re-using the proof of [GHJY15] with the observation that local optima are preserved by linear transformation. The intermediate regime when $d < n < d^{1+\varepsilon}$ should be analyzable by Kac-Rice formula using similar techniques, but our current proof cannot capture it directly. Since the proof in this paper is already involved, we leave this case to future work. The condition $n < d^{2-\varepsilon}$ matches the best over-completeness level that existing polynomial algorithm can handle [DLCC07, MSS16].

**Our techniques** The proof of Theorem 1.1 uses Kac-Rice formula (see, e.g., [AT09]), which is based on a counting argument. To build up the intuition, we tentatively view the unit sphere as a collection of discrete points, then for each point $x$ one can compute the probability (with respect to the randomness of the function) that $x$ is a local maximum. Adding up all these probabilities will give us the expected number of local maxima. In continuous space, such counting argument has to be more delicate since the local geometry needs to be taken into account. This is formalized by Kac-Rice formula (see Lemma 2.2).

However, Kac-Rice formula only gives a closed form expression that involves the integration of the expectation of some complicated random variable. It's often very challenging to simplify the expression to obtain interpretable results. Before our work, Auffinger et al. [AAČ13, AA+13] have successfully applied Kac-Rice formula to characterize the landscape of polynomials with random Gaussian coefficients. The exact expectation of the number of local minima can be computed there, because the Hessian of a random polynomial is a Gaussian orthogonal ensemble, whose eigenvalue distribution is well-understood with closed form expression.

Our technical contribution here is successfully applying Kac-Rice formula to *structured* random non-convex functions where the formula cannot be exactly evaluated. The Hessian and gradients of $f(\cdot)$ have much more complicated distributions compared to the Gaussian orthogonal ensemble. As a result, the Kac-Rice formula is difficult to be evaluated exactly. We instead cut the space $\mathbb{R}^d$ into regions and use different techniques to *estimate* the number of local maxima. See a proof overview in Section 3. We believe our techniques can be extended to 3rd order tensors and can shed light on the analysis of other non-convex problems with structured randomness.

**Organization** In Section 2 we introduce preliminaries regarding manifold optimization and Kac-Rice formula. We give a detailed explanation of our proof strategy in Section 3. The technical details are deferred to the supplementary material. We also note that the supplementary material contains an extended version of the preliminary and proof overview section below.

## 2 Notations and Preliminaries

We use $\mathrm{Id}_d$ to denote the identity matrix of dimension $d \times d$. Let $\| \cdot \|$ denote the spectral norm of a matrix or the Euclidean norm of a vector. Let $\|\cdot\|_F$ denote the Frobenius norm of a matrix or a tensor.

**Gradient, Hessian, and local maxima on manifold** We have a constrained optimization problem over the unit sphere $S^{d-1}$, which is a smooth manifold. Thus we define the local maxima with respect to the manifold. It's known that projected gradient descent for $S^{d-1}$ behaves pretty much the same on the manifold as in the usual unconstrained setting [BAC16]. In supplementary material we give a brief introduction to manifold optimization, and the definition of gradient and Hessian. We refer the readers to the book [AMS07] for more backgrounds.

Here we use grad $f$ and Hess $f$ to denote the gradient and the Hessian of $f$ on the manifold $S^{d-1}$. We compute them in the following claim.

**Claim 2.1.** *Let $f : S^{d-1} \to \mathbb{R}$ be $f(x) := \frac{1}{4} \sum_{i=1}^{n} \langle a_i, x \rangle^4$. Let $P_x = \mathrm{Id}_d - xx^\top$. Then the gradient and Hessian of $f$ on the sphere can be written as,*

$$\mathrm{grad}\, f(x) = P_x \sum_{i=1}^{n} \langle a_i, x \rangle^3 a_i , \;\; \mathrm{Hess}\, f(x) = 3 \sum_{i=1}^{n} \langle a_i, x \rangle^2 P_x a_i a_i^\top P_x - \left( \sum_{i=1}^{n} \langle a_i, x \rangle^4 \right) P_x ,$$

A local maximum of a function $f$ on the manifold $S^{d-1}$ satisfies grad $f(x) = 0$, and Hess $f(x) \preceq 0$. Let $\mathcal{M}_f$ be the set of all local maxima, i.e. $\mathcal{M}_f = \left\{ x \in S^{d-1} : \mathrm{grad}\, f(x) = 0, \mathrm{Hess}\, f(x) \preceq 0 \right\}$.

**Kac-Rice formula** Kac-Rice formula is a general tool for computing the expected number of special points on a manifold. Suppose there are two random functions $P(\cdot) : \mathbb{R}^d \to \mathbb{R}^d$ and $Q(\cdot) : \mathbb{R}^d \to \mathbb{R}^k$, and an open set $\mathcal{B}$ in $\mathbb{R}^k$. The formula counts the expected number of point $x \in \mathbb{R}^d$ that satisfies both $P(x) = 0$ and $Q(x) \in \mathcal{B}$.

Suppose we take $P = \nabla f$ and $Q = \nabla^2 f$, and let $\mathcal{B}$ be the set of negative semidefinite matrices, then the set of points that satisfies $P(x) = 0$ and $Q \in \mathcal{B}$ is the set of all local maxima $\mathcal{M}_f$. Moreover, for any set $Z \subset S^{d-1}$, we can also augment $Q$ by $Q = [\nabla^2 f, x]$ and choose $\mathcal{B} = \{A : A \preceq 0\} \otimes Z$.

With this choice of $P, Q$, Kac-Rice formula can count the number of local maxima inside the region $Z$. For simplicity, we will only introduce Kac-Rice formula for this setting. We refer the readers to [AT09, Chapter 11&12] for more backgrounds.

**Lemma 2.2** (Informally stated). *Let $f$ be a random function defined on the unit sphere $S^{d-1}$ and let $Z \subset S^{d-1}$. Under certain regularity conditions[3] on $f$ and $Z$, we have*

$$\mathbb{E}\left[|\mathcal{M}_f \cap Z|\right] = \int_x \mathbb{E}\left[|\det(\text{Hess } f)| \cdot \mathbf{1}(\text{Hess } f \preceq 0)\mathbf{1}(x \in Z) \mid \text{grad } f(x) = 0\right] p_{\text{grad } f(x)}(0)dx. \quad (2.1)$$

*where $dx$ is the usual surface measure on $S^{d-1}$ and $p_{\text{grad } f(x)}(0)$ is the density of $\text{grad } f(x)$ at 0.*

**Formula for the number of local maxima** In this subsection, we give a concrete formula for the number of local maxima of our objective function (1.1) inside the superlevel set $L$ (defined in equation (1.2)). Taking $Z = L$ in Lemma 2.2, it boils down to estimating the quantity on the right hand side of (2.1). We remark that for the particular function $f$ as defined in (1.1) and $Z = L$, the integrand in (2.1) doesn't depend on the choice of $x$. This is because for any $x \in S^{d-1}$, $(\text{Hess } f, \text{grad } f, \mathbf{1}(x \in L))$ has the same joint distribution, as characterized below:

**Lemma 2.3.** *Let $f$ be the random function defined in (1.1). Let $\alpha_1, \ldots, \alpha_n \in \mathcal{N}(0,1)$, and $b_1, \ldots, b_n \sim \mathcal{N}(0, \text{Id}_{d-1})$ be independent Gaussian random variables. Let*

$$M = \|\alpha\|_4^4 \cdot \text{Id}_{d-1} - 3\sum_{i=1}^n \alpha_i^2 b_i b_i^\top \text{ and } g = \sum_{i=1}^n \alpha_i^3 b_i \quad (2.2)$$

*Then, we have that for any $x \in S^{d-1}$, $(\text{Hess } f, \text{grad } f, f)$ has the same joint distribution as $(-M, g, \|\alpha\|_4^4)$.*

Using Lemma 2.2 (with $Z = L$) and Lemma 2.3, we derive the following formula for the expectation of our random variable $\mathbb{E}\left[|\mathcal{M}_f \cap L|\right]$. Later we will later use Lemma 2.2 slightly differently with another choice of $Z$.

**Lemma 2.4.** *Using the notation of Lemma 2.3, let $p_g(\cdot)$ denote the density of $g$. Then,*

$$\mathbb{E}\left[|\mathcal{M}_f \cap L|\right] = \text{Vol}(S^{d-1}) \cdot \mathbb{E}\left[|\det(M)| \mathbf{1}(M \succeq 0)\mathbf{1}(\|\alpha\|_4^4 \geq 3(1+\zeta)n) \mid g = 0\right] p_g(0). \quad (2.3)$$

## 3 Proof Overview

In this section, we give a high-level overview of the proof of the main Theorem. We will prove a slightly stronger version of Theorem 1.1.

Let $\gamma$ be a universal constant that is to be determined later. Define the set $L_1 \subset S^{d-1}$ as,

$$L_1 := \left\{ x \in S^{d-1} : \sum_{i=1}^n \langle a_i, x \rangle^4 \geq 3n + \gamma\sqrt{nd} \right\}. \quad (3.1)$$

Indeed we see that $L$ (defined in (1.2)) is a subset of $L_1$ when $n \gg d$. We prove that in $L_1$ there are exactly $2n$ local maxima.

**Theorem 3.1** (main). *There exists universal constants $\gamma, \beta$ such that the following holds: suppose $d^2/\log^{O(1)} \geq n \geq \beta d \log^2 d$ and $L_1$ be defined as in (3.1), then with high probability over the choice of $a_1, \ldots, a_n$, we have that the number of local maxima in $L_1$ is exactly $2n$:*

$$|\mathcal{M}_f \cap L_1| = 2n. \quad (3.2)$$

*Moreover, each of the local maximum in $L_1$ is $\widetilde{O}(\sqrt{n/d^3})$-close to one of $\pm\frac{1}{\sqrt{d}}a_1, \ldots, \pm\frac{1}{\sqrt{d}}a_n$.*

In order to count the number of local maxima in $L_1$, we use the Kac-Rice formula (Lemma 2.4). Recall that what Kac-Rice formula gives an expression that involves the complicated expectation

$\mathbb{E}\left[|\det(M)| \, \mathbf{1}(M \succeq 0)\mathbf{1}(\|\alpha\|_4^4 \geq 3(1+\zeta)n) \mid g = 0\right]$. Here the difficulty is to deal with the determinant of a random matrix $M$ (defined in Lemma 2.3), whose eigenvalue distribution does not admit an analytical form. Moreover, due to the existence of the conditioning and the indicator functions, it's almost impossible to compute the RHS of the Kac-Rice formula (equation (2.3)) exactly.

**Local vs. global analysis:** The key idea to proceed is to divide the superlevel set $L_1$ into two subsets

$$L_1 = (L_1 \cap L_2) \cup L_2^c,$$

$$\text{where } L_2 := \left\{x \in S^{d-1} : \forall i, \|P_x a_i\|^2 \geq (1-\delta)d, \text{ and } |\langle a_i, x\rangle|^2 \leq \delta d\right\}. \quad (3.3)$$

Here $\delta$ is a sufficiently small universal constant that is to be chosen later. We also note that $L_2^c \subset L_1$ and hence $L_1 = (L_1 \cap L_2) \cup L_2^c$.

Intuitively, the set $L_1 \cap L_2$ contains those points that do not have large correlation with any of the $a_i$'s; the compliment $L_2^c$ is the union of the neighborhoods around each of the desired vector $\frac{1}{\sqrt{d}}a_1, \ldots, \frac{1}{\sqrt{d}}a_n$. We will refer to the first subset $L_1 \cap L_2$ as the global region, and refer to the $L_2^c$ as the local region.

We will compute the number of local maxima in sets $L_1 \cap L_2$ and $L_2^c$ separately using different techniques. We will show that with high probability $L_1 \cap L_2$ contains no local maxima using Kac-Rice formula (see Theorem 3.2). Then, we show that $L_2^c$ contains exactly $2n$ local maxima (see Theorem 3.3) using a different and more direct approach.

**Global analysis.** The key benefit of have such division to local and global regions is that for the global region, we can avoid evaluating the value of the RHS of the Kac-Rice formula. Instead, we only need to have an *estimate*: Note that the number of local optima in $L_1 \cap L_2$, namely $|\mathcal{M}_f \cap L_1 \cap L_2|$, is an integer nonnegative random variable. Thus, if we can show its expectation $\mathbb{E}\left[|\mathcal{M}_f \cap L_1 \cap L_2|\right]$ is much smaller than 1, then Markov's inequality implies that with high probability, the number of local maxima will be *exactly* zero. Concretely, we will use Lemma 2.2 with $Z = L_1 \cap L_2$, and then estimate the resulting integral using various techniques in random matrix theory. It remains quite challenging even if we are only shooting for an estimate. Concretely, we get the following Theorem

**Theorem 3.2.** *Let sets $L_1, L_2$ be defined as in equation (3.3) and $n \geq \beta d \log^2 d$. There exists universal small constant $\delta \in (0,1)$ and universal constants $\gamma, \beta$, and a high probability event $G_0$, such that the expected number of local maxima in $L_1 \cap L_2$ conditioned on $G_0$ is exponentially small:*

$$\mathbb{E}\left[|\mathcal{M}_f \cap L_1 \cap L_2| \mid G_0\right] \leq 2^{-d/2}.$$

See Section 3.1 for an overview of the analysis. The purpose and definition of $G_0$ are more technical and can be found in Section 3 of the supplementary material around equation (3.3) (3,4) and (3.5). We also prove that $G_0$ is indeed a high probability event in supplementary material. [4]

**Local analysis.** In the local region $L_2^c$, that is, the neighborhoods of $a_1, \ldots, a_n$, we will show there are *exactly* $2n$ local maxima. As argued above, it's almost impossible to get exact numbers out of the Kac-Rice formula since it's often hard to compute the complicated integral. Moreover, Kac-Rice formula only gives the expected number but not high probability bounds. However, here the observation is that the local maxima (and critical points) in the local region are well-structured. Thus, instead, we show that in these local regions, the gradient and Hessian of a point $x$ are dominated by the terms corresponding to components $\{a_i\}$'s that are highly correlated with $x$. The number of such terms cannot be very large (by restricted isometry property, see Section B.5 of the supplementary material). As a result, we can characterize the possible local maxima explicitly, and eventually show there is exactly one local maximum in each of the local neighborhoods around $\{\pm\frac{1}{\sqrt{d}}a_i\}$'s. Similar (but weaker) analysis was done before in [AGJ15]. We formalize the guarantee for local regions in the following theorem, which is proved in Section 5 of the supplementary material. In Section 3.2 of the supplementary material, we also discuss the key ideas of the proof of this Theorem.

**Theorem 3.3.** *Suppose $1/\delta^2 \cdot d \log d \leq n \leq d^2 / \log^{O(1)} d$. Then, with high probability over the choice $a_1, \ldots, a_n$, we have,*

$$|\mathcal{M}_f \cap L_1 \cap L_2^c| = 2n. \quad (3.4)$$

*Moreover, each of the point in $L \cap L_2^c$ is $\widetilde{O}(\sqrt{n/d^3})$-close to one of $\pm\frac{1}{\sqrt{d}}a_1, \ldots, \pm\frac{1}{\sqrt{d}}a_n$.*

The main Theorem 3.1 is a direct consequence of Theorem 3.2 and Theorem 3.3. The formal proof can be found in Section 3 of the supplementary material.

In the next subsections we sketch the basic ideas behind the proof of Theorem 3.2 and Theorem 3.3. Theorem 3.2 is the crux of the technical part of the paper.

## 3.1   Estimating the Kac-Rice formula for the global region

The general plan to prove Theorem 3.2 is to use random matrix theory to estimate the RHS of the Kac-Rice formula. We begin by applying Kac-Rice formula to our situation. We note that we dropped the effect of $G_0$ in all of the following discussions since $G_0$ only affects some technicality that appears in the details of the proof in the supplementary material.

**Applying Kac-Rice formula.**   The first step to apply Kac-Rice formula is to characterize the joint distribution of the gradient and the Hessian. We use the notation of Lemma 2.3 for expressing the joint distribution of $(\text{Hess } f, \text{grad } f, \mathbf{1}(x \in L_1 \cap L_2))$. For any fix $x \in S^{d-1}$, let $\alpha_i = \langle a_i, x \rangle$ and $b_i = P_x a_i$ (where $P_x = \text{Id} - xx^\top$) and $M = \|\alpha\|_4^4 \cdot \text{Id}_{d-1} - 3\sum_{i=1}^n \alpha_i^2 b_i b_i^\top$ and $g = \sum_{i=1}^n \alpha_i^3 b_i$ as defined in (2.2). In order to apply Kac-Rice formula, we'd like to compute the joint distribution of the gradient and the Hessian. We have that $(\text{Hess } f, \text{grad } f, \mathbf{1}(x \in L_1 \cap L_2))$ has the same distribution as $(M, g, \mathbf{1}(E_1 \cap E_2 \cap E_2'))$, where $E_1$ corresponds to the event that $x \in L_1$,

$$E_1 = \left\{ \|\alpha\|_4^4 \geq 3n + \gamma\sqrt{nd} \right\},$$

and events $E_2$ and $E_2'$ correspond to the events that $x \in L_2$. We separate them out to reflect that $E_2$ and $E_2'$ depends the randomness of $\alpha_i$'s and $b_i$'s respectively.

$$E_2 = \left\{ \|\alpha\|_\infty^2 \leq \delta d \right\}, \text{ and } E_2' = \left\{ \forall i \in [n], \|b_i\|^2 \geq (1-\delta)d \right\}.$$

Using Kac-Rice formula (Lemma 2.2 with $Z = L_1 \cap L_2$), we conclude that

$$\mathbb{E}\left[|\mathcal{M}_f \cap L_1 \cap L_2|\right] = \text{Vol}(S^{d-1}) \cdot \mathbb{E}\left[|\det(M)| \mathbf{1}(M \succeq 0)\mathbf{1}(E_1 \cap E_2 \cap E_2') \mid g = 0\right] p_g(0). \tag{3.5}$$

Next, towards proving Theorem 3.2 we will estimate the RHS of (3.5) using various techniques.

**Conditioning on $\alpha$.** We observe that the distributions of the gradient $g$ and Hessian $M$ on the RHS of equation 3.5 are fairly complicated. In particular, we need to deal with the interactions of $\alpha_i$'s (the components along $x$) and $b_i$'s (the components in the orthogonal subspace of $x$). Therefore, we use the law of total expectation to first condition on $\alpha$ and take expectation over the randomness of $b_i$'s, and then take expectation over $\alpha_i$'s. Let $p_{g|\alpha}$ denotes the density of $g \mid \alpha$, using the law of total expectation, we have,

$$\mathbb{E}\left[|\det(M)| \mathbf{1}(M \succeq 0)\mathbf{1}(E_1 \cap E_2 \cap E_2') \mid g = 0\right] p_g(0)$$
$$= \mathbb{E}\left[\mathbb{E}\left[|\det(M)| \mathbf{1}(M \succeq 0)\mathbf{1}(E_2') \mid g = 0, \alpha\right] \mathbf{1}(E_1)\mathbf{1}(E_2)p_{g|\alpha}(0)\right]. \tag{3.6}$$

Note that the inner expectation of RHS of (3.6) is with respect to the randomness of $b_i$'s and the outer one is with respect to $\alpha_i$'s.

For notional convenience we define $h(\cdot) : \mathbb{R}^n \to \mathbb{R}$ as

$$h(\alpha) := \text{Vol}(S^{d-1}) \mathbb{E}\left[\det(M)\mathbf{1}(M \succeq 0)\mathbf{1}(E_2') \mid g = 0, \alpha\right] \mathbf{1}(E_1)\mathbf{1}(E_2)p_{g|\alpha}(0).$$

Then, using the Kac-Rice formula (equation (2.3))[5] and equation (3.5), we obtain the following explicit formula for the number of local maxima in $L_1 \cap L_2$.

$$\mathbb{E}\left[|\mathcal{M}_f \cap L_1 \cap L_2|\right] = \mathbb{E}\left[h(\alpha)\right]. \tag{3.7}$$

We note that $p_{g|\alpha}(0)$ has an explicit expression since $g \mid \alpha$ is Gaussian. For the ease of exposition, we separate out the hard-to-estimate part from $h(\alpha)$, which we call $W(\alpha)$:

$$W(\alpha) := \mathbb{E}\left[\det(M)\mathbf{1}(M \succeq 0)\mathbf{1}(E_2') \mid g = 0, \alpha\right] \mathbf{1}(E_1)\mathbf{1}(E_2). \tag{3.8}$$

Therefore by definition, we have that $h(\alpha) = \text{Vol}(S^{d-1})W(\alpha)p_{g|\alpha}(0)$. Now, since we have conditioned on $\alpha$, the distributions of the Hessian, namely $M \mid \alpha$, is a generalized Wishart matrix which is slightly easier than before. However there are still several challenges that we need to address in order to estimate $W(\alpha)$.

**How to control** $\det(M)\mathbf{1}(M \succeq 0)$**?** Recall that $M = \|\alpha\|_4^4 - 3\sum \alpha_i^2 b_i b_i^\top$, which is a generalized Wishart matrix whose eigenvalue distribution has no (known) analytical expression. The determinant itself by definition is a high-degree polynomial over the entries, and in our case, a complicated polynomial over the random variables $\alpha_i$'s and vectors $b_i$'s. We also need to properly exploit the presence of the indicator function $\mathbf{1}(M \succeq 0)$, since otherwise, the desired statement will not be true – the function $f$ has an exponential number of critical points.

Fortunately, in most of the cases, we can use the following simple claim that bounds the determinant from above by the trace. The inequality is close to being tight when all the eigenvalues of $M$ are similar to each other. More importantly, it uses naturally the indicator function $\mathbf{1}(M \succeq 0)$! Later we will see how to strengthen it when it's far from tight.

**Claim 3.4.** *We have that*

$$\det(M)\mathbf{1}(M \succeq 0) \leq \left( \frac{|\text{tr}(M)|}{d-1} \right)^{d-1} \mathbf{1}(M \succeq 0)$$

The claim is a direct consequence of AM-GM inequality on the eigenvalue of $M$. (Note that $M$ is of dimension $(d-1) \times (d-1)$. we give a formal proof in Section 3.1 of the supplementary material). It follows that

$$W(\alpha) \leq \mathbb{E}\left[ \frac{|\text{tr}(M)|^{d-1}}{(d-1)^{d-1}} \mid g = 0, \alpha \right] \mathbf{1}(E_1). \tag{3.9}$$

Here we dropped the indicators for events $E_2$ and $E_2'$ since they are not important for the discussion below. It turns out that $|\text{tr}(M)|$ is a random variable that concentrates very well, and thus we have $\mathbb{E}\left[ |\text{tr}(M)|^{d-1} \right] \approx |\mathbb{E}\left[\text{tr}(M)\right]|^{d-1}$. It can be shown that (see Proposition 4.3 in the supplementary material for the detailed calculation),

$$\mathbb{E}\left[\text{tr}(M) \mid g = 0, \alpha\right] = (d-1)\left( \|\alpha\|_4^4 - 3\|\alpha\|^2 + 3\|\alpha\|_8^8/\|\alpha\|_6^6 \right).$$

Therefore using equation (3.9) and equation above, we have that

$$W(\alpha) \leq \left( \|\alpha\|_4^4 - 3\|\alpha\|^2 + 3\|\alpha\|_8^8/\|\alpha\|_6^6 \right)^{d-1} \mathbf{1}(E_0)\mathbf{1}(E_1).$$

Note that since $g \mid \alpha$ has Gaussian distribution, we have, $p_{g|\alpha}(0) = (2\pi)^{-d/2}(\|\alpha\|_6^6)^{-d/2}$. Thus using two equations above, we can bound $\mathbb{E}\left[h(\alpha)\right]$ by

$$\mathbb{E}\left[h(\alpha)\right] \leq \text{Vol}(S^{d-1})\mathbb{E}\left[ \left( \|\alpha\|_4^4 - 3\|\alpha\|^2 + 3\|\alpha\|_8^8/\|\alpha\|_6^6 \right)^{d-1} \cdot (2\pi)^{-d/2}(\|\alpha\|_6^6)^{-d/2}\mathbf{1}(E_0)\mathbf{1}(E_1) \right].$$
$$\tag{3.10}$$

Therefore, it suffices to control the RHS of (3.10), which is much easier than the original Kac-Rice formula. However, it turns out that RHS of (3.10) is roughly $c^d$ for some constant $c > 1$! Roughly speaking, this is because the high powers of a random variables is very sensitive to its tail.

**Two sub-cases according to** $\max|\alpha_i|$**.** We aim to find a tighter bond of $\mathbb{E}[h(\alpha)]$ by re-using the idea in equation (3.10). Intuitively we can consider two separate situations events: the event $F_0$ when all of the $\alpha_i$'s are close to constant and the complementary event $F_0^c$. Formally, let $\tau = Kn/d$ where $K$ is a universal constant that will be determined later. Let $F_0$ be the event that .$F_0 = \left\{ \|\alpha\|_\infty^4 \leq \tau \right\}$. Then we control $\mathbb{E}\left[h(\alpha)\mathbf{1}(F_0)\right]$ and $\mathbb{E}\left[h(\alpha)\mathbf{1}(F_0^c)\right]$ separately. For the former, we basically need to reuse the equation (3.10) with an indicator function inserted inside the expectation. For the latter, we make use of the large coordinate, which contributes to the $-3\alpha_i^2 b_i b_i^\top$ term in $M$ and makes the probability of $\mathbf{1}(M \succeq 0)$ extremely small. As a result $\det(M)\mathbf{1}(M \succeq 0)$ is almost always 0. We formalized the two cases as below:

**Proposition 3.5.** *Let $K \geq 2 \cdot 10^3$ be a universal constant. Let $\tau = Kn/d$ and let $\gamma, \beta$ be sufficiently large constants (depending on $K$). Then for any $n \geq \beta d \log^2 d$, we have that*

$$\mathbb{E}\left[h(\alpha)\mathbf{1}(F_0)\right] \leq (0.3)^{d/2}.$$

**Proposition 3.6.** *In the setting of Proposition 3.5, we have*

$$\mathbb{E}\left[h(\alpha)\mathbf{1}(F_0^c)\right] \le n \cdot (0.3)^{d/2}.$$

We see that Theorem 3.2 can be obtained as a direct consequence of Proposition 3.5, Proposition 3.6 and equation (3.7).

Due to space limit, we refer the readers to the supplementary material for an extended version of proof overview and the full proofs.

# 4   Conclusion

We analyze the optimization landscape of the random over-complete tensor decomposition problem using the Kac-Rice formula and random matrix theory. We show that in the superlevel set $L$ that contains all the points with function values barely larger than the random guess, there are exactly $2n$ local maxima that correspond to the true components. This implies that with an initialization slight better than the random guess, local search algorithms converge to the desired solutions. We believe our techniques can be extended to 3rd order tensors, or other non-convex problems with structured randomness.

The immediate open question is whether there is any other spurious local maximum outside this superlevel set. Answering it seems to involve solving difficult questions in random matrix theory. Another potential approach to unravel the mystery behind the success of the non-convex methods is to analyze the early stage of local search algorithms and show that they will enter the superlevel set $L$ quickly from a good initialization.

## Footnotes

[1]Power method is exactly equivalent to gradient ascent with a properly chosen finite learning rate

[2]We note that by stochastic gradient descent we meant the algorithm that is analyzed in [GHJY15]. To get a global maximum in polynomial time (polynomial in $\log(1/\varepsilon)$ to get $\varepsilon$ precision), one also needs to slightly modify stochastic gradient descent in the following way: run SGD until $1/d$ accuracy and then switch to gradient descent. Since the problem is locally strongly convex, the local convergence is linear.

[3]We omit the long list of regularity conditions here for simplicity. See more details at [AT09, Theorem 12.1.1]

[4]We note again that the supplementary material contains more details in each section even for sections in the main text.

[5]In Section C of the supplementary material, we rigorously verify the regularity condition of Kac-Rice formula.

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
