[Reviews · NeurIPS 2017]

Reviewer 1



### Summary The paper analyzes the landscape of tensor decomposition problem. Specifically, it studies random over-complete tensors. The associated objective function is nonconvex, yet in practice simple methods based on gradient ascent are observed to solve this problem. This paper proves why we should expect such outcome by showing that there is almost no local maxima other than the global maxima of the problem when the optimization is initialized by any solution that is slightly better than random guess. Importantly, it is shown that these initial points do not have to be close to the true components of the tensor. This is an interesting result and well written paper. The analysis involves two steps: local (points close to true components) and global (point far from true components). The number of local maxima in each case is analyzed and shown to be exactly 2n for the former and almost nonexistent for the latter. ### Questions 1. Authors mention that in practice gradient ascent type algorithms succeed. Do these algorithms use any specific initialization that meets your criterion? More precisely, do they use a scheme to produce initialization which is slightly better than chance and thus the objective value is slightly greater than 3*n? IF YES: Please elaborate briefly how this is done or provide a reference. IF NO: Any idea how such initial points could be generated easily? Could it be a very hard problem on its own? 2. The analysis is performed on random tensors, where each true component is assumed to be drawn iid from Gaussian distribution N (0, I). It seems to be a strong assumption and not true for real world problems. While I am not disputing the importance of the result provided in this paper, I was wondering if the authors had thoughts about analysis scenarios that could potentially explain the success of local schemes in a broader setup? For example, performing a worst-case analysis (i.e. choosing the worst tensors) and then showing that these worst tensors form a negligible fraction of the space of tensors? This would go beyond the Gaussian setting and may be more pertinent to explaining why local search succeeds on real world problems. ### Minor Comments 1. Lemma 2.2. Eq. 2.1, the first E is missing a | at the begining right after [. 2. Lines 80, 84, 123 local minima should be local maxima? 3. You refer to hard optimization as nonconvex, but since your task in 1.1. is defined as a maximization, concave functions are actually ideal for that. To avoid confusion around minimization/maximization/nonconvexity, you might want to replace 1.1 from Max f(x) with Min -f(x) so that everything remains a minimization, and thus nonconvexity implies hardness. In that case, you can also ignore my comment 2.

Reviewer 2



This paper addresses the problem of identifying the local/global maxima for tensor decomposition. Tensor decomposition is an increasingly important non-convex problem with applications to latent variable models and neural networks. One of the fundamental question in tensor decomposition is, when are the components identifiable? and is there efficient methods for finding those components. It has been noted that LOCALLY the geometry of this landscape is close to convex and a simple methods such as projected power method with good initializations can identify those components. The major gap in this picture is that, empirically randomly initialized methods seem to find those components just as well. This paper is an important step towards filling this gap. A breakthrough made in this paper is in bringing the Kac-Rice formula to fill in the missing piece in the puzzle. How is the local minima behaving GLOBALLY, that is away from any of the components? This is made precise by counting argument based on the expectation of the number of such minima. There are no such minima (with some technical conditions on the region) with high probability. This is an exciting new result, with minor weaknesses on 1) it is for Gaussian components 2) it is for symmetric 4-th order tensor only. The theorem provides a guarantee in the over complete regime where n \geq d \log d. Is there something fundamentally different between over and under complete regime from the perspective of the geometry of the objective function? One would expect under completed tensor to be easier to decompose, and hence at least the same guarantee as in the main theorem to hold. Similarly, the main result holds up to n \leq d^2/log d. Is this a fundamental limit on how many components can be learned? How does this compare to other (weaker) results known for tensor decomposition? For example, what are the conditions sufficient for simpler LOCAL convexity?